

# Wind variability in the Canary Current during the last 70 years

Nerea Marrero Betancort[1], Javier Marcello[1], Dionisio Rodríguez Esparragón[1], Santiago Hernández-León[2]

[1]Grupo de Procesado de Imágenes y Teledetección, Instituto de Oceanografía y Cambio Global (IOCAG), Universidad de Las Palmas de Gran Canaria, Unidad Asociada ULPGC-CSIC, Las Palmas de Gran Canaria, 35017, Spain
[2]Grupo de Oceanografía Biológica, Instituto de Oceanografía y Cambio Global (IOCAG), Universidad de Las Palmas de Gran Canaria, Unidad Asociada ULPGC-CSIC, Las Palmas de Gran Canaria, 35017, Spain

*Correspondence*: Nerea Marrero Betancort (nerea.marrero102@alu.ulpgc.es)

**Abstract.** Climate evolves following natural variability and knowledge of these trends is of paramount importance to understand future scenarios in the frame of global change. Obtaining local data is also of importance since climatic anomalies depend on the geographical area. In this sense, the Canary Current is located in one of the major eastern boundary current systems of the oceans and is mainly driven by the Trade Winds. The latter promotes Ekman transport and give rise to one of the most important upwelling zones of the word in the Northwest African coast. Nearly 30 years ago, Bakun raised a hypothesis contending that coastal upwelling in eastern boundary upwelling systems (EBUS) might be intensified as the effect of global warming due to the enhancement of the Trade Winds as the effect of increasing pressure differences between the ocean and the continent. Using available NCEP/NCAR wind data north of the Canary Islands from 1948 to 2017, we show that Trade Winds intensity experienced a net decrease of 1 m·s[-1]. Moreover, theses winds are strongly influenced, as expected, by the large-scale atmospheric patterns, such as the North Atlantic Oscillation (NAO). In addition, we found a relationship between the wind pattern and the Atlantic Multidecadal Oscillation (AMO), indicating that the ocean contributes to the multidecadal atmospheric variability in this area of the ocean with a considerable lag (>10 years).

## 1 Introduction

The just-released Intergovernmental Panel on Climate Change (IPCC) special report on the ocean and cryosphere in a changing climate (SROCC) (IPCC, 2019) details the immense pressure that climate change is exerting on ocean ecosystems and portrays a disastrous future for most life in the ocean and for the billions of people who depend on it. One of these ocean ecosystems of paramount importance for fisheries are the Eastern Boundary Current Systems (EBCS). The four major EBCS of the world ocean are the California, Canary, Benguela, and Humboldt systems (Bakun and Nelson, 1991). The EBCS cover a surface area of only ~2% of the global oceans but produce about 20% of the world fisheries (Pauly and Christensen, 1995). Currently, the research about the response of the EBCS and the associated impact under a global climate change scenario have motivated numerous studies over different time periods. These recent analyses covering the variability of physical,





geological, biological, and chemical characteristics in EBCS in relation to global climate change are rather controversial because of the different results (Pardo et al., 2011).

A long-standing hypothesis contends that coastal upwelling in EBCS might be intensified as the effect of global warming (Bakun, 1990). Trade Winds could be intensified because of the increase in the pressure gradient between the continents and the ocean, promoting an increase in Ekman pumping in the coastal zone. Thus, as global warming progresses it is expected
an increase of Trade Wind intensity in those areas. These changes in the ocean physics would provide more nutrients and therefore primary productivity fuelling upper trophic levels such as an increase in fish crop.

In this context, several studies reveal an increase in the strengthening of upwelling, in the main Eastern Boundaries Upwelling Systems (EBUS) of the world such as California, Humboldt (Chile-Peru), Canary, and Benguela (Demarcq, 2009; Gutiérrez et al., 2011; Lima and Wethey, 2012; Santos et al., 2012a,b; Cropper et al., 2014; Sydeman et al., 2014; Benazzouz
et al., 2015; Varela et al. 2015). Sydeman et al. (2014) performed a meta-analysis of the literature on upwelling-favourable wind intensification, with results from 22 studies published between 1990 and 2012 and based on time series ranging in duration from 17 to 61 years. This research illustrated that winds have intensified in the California, Benguela, and Humboldt upwelling systems. However, wind change is equivocal in the Canary system. In this sense, the 40-year study (1967-2017) performed by Barton et al. (2013) analysing the atmospheric and oceanic variables involved in the Canary EBUS showed a
lack of statistically significant change in the meridional wind component (upwelling favourable).

Recently, Bonino et al. (2019) found sharp differences in Atlantic and Pacific upwelling areas, highlighting the uniqueness of each EBUS and observing a negative upwelling trend connected to the low frequency of the Atlantic Multidecadal Oscillation (AMO) in the Canary Current System. Therefore, the negative trend observed could be the result of the long-term variability of the AMO index.

Thus, uncertainty about the behaviour of the wind patterns in the Canary Current promotes considerable concern in vulnerable areas such as the Canary Islands due to their expected alterations due to climate change (Kelman and Khan, 2013; Nurse et al., 2014). Therefore, we have analysed changes in wind patterns during the last 70 years (1948-2017) in this archipelago in order to explore the connections with future climate change and the North Atlantic global circulation patterns.

## 2 Material and Methods

### 2.1 Study Area

In order to study long-term trends in wind direction and intensity in the Canary Islands, we selected a location to the north of the archipelago (29.52˚N, 15˚W, Fig. 1), away from the coast to avoid disturbances due to island orography and close to the European Oceanic Time Series Station of the Canary Islands (ESTOC, Neuer et al., 2007).





## 2.2 Data

The wind data used in this study was produced by the National Atmospheric Prediction Center and the United States National Atmospheric Research Center (NCEP/NCAR) and provided by the Physical Sciences Division (PSD) of the National Oceanic and Atmospheric Administration (NOAA) (Kalnay et al., 1996). These data correspond to a reanalysis of the monthly means of the zonal (u) and meridional (v) components of the wind, measured at 10 m height. It is supplied in a matrix of 192 pixels in longitude and 94 pixels in latitude (Gaussian grid), and a geographical resolution of 2.5° x 2.5°. For

our study, the time period between January 1948 and December 2017 was chosen, covering a total of 70 years.

Additionally, data derived from the Cross-Calibrated Multi-Platform (CCMP) project was also analysed, combining a series of calibrated satellite wind data obtained by remote sensors, supplied by the Goddard Space Flight Center (GSFC, NASA), and disseminated by the Physical Oceanography Distributed Active Archive Center (PODAAC) (Atlas et al., 2011). PODAAC data consist of monthly averages of the zonal and meridional components of the wind, also measured at 10 m

height. It is supplied in a matrix of 1440 pixels in longitude and 628 pixels in latitude, with a geographical resolution of 0.25° x 0.25°. These data have a temporal range lower than the NCEP, spanning a period of time from January 1988 to December 2011, a total of 24 years

## 2.3 Methodology

The NCEP and PODAAC data used have a level 4 processing and a preliminary filter was applied to eliminate anomalous

data such as outliers and negative velocity data values. Next, we extracted the subset of wind data of interest for this study corresponding to the area between 27° N and 30° N, and 11° W to 20° W. Thereafter, wind intensity and direction were calculated from the zonal and meridional wind components. Subsequently, we performed different correlation analysis between PODAAC and NCEP data. As a result of the high correlation achieved between both datasets, the time period between January 1948 and December 2017 was chosen in order to cover the range of 70 years that allows a more complete

wind analysis. The oceanographic standard was used to describe the wind, that is, the direction towards which the wind blows, taking as reference the geographical north (0°) and measuring the angle clockwise.

## 3. Results

In order to select the most appropriate dataset for this study, as indicated, we carried out a preliminary statistical analysis of

the correlation between the NCEP data and the PODAAC series. The study covered the 24 years period in which data are available from both sources (from 1988 to 2011). Both datasets were highly correlated, both in direction (r=0.949, Fig. 2a) and intensity (r=0.923, Fig. 2b). According to these results, we decided to choose the NCEP data for the study because the time range is considerably longer.





Time-series of wind direction showed a linear regression with a slope of 0.013º for the entire period, which implied a total
variation in direction of 10.7º after the 70 years. Thus, the wind direction during the complete period of study varied slightly,
with a net trend to rotate clockwise. The average wind direction value was 197º ± 35.5° SD, which is the usual direction of
the Trade Winds in the study area. The most remarkable anomalies occurred in autumn and winter being 338˚, 15˚, and 11˚
in February 1965, January 1969, and December 2001, respectively (Fig. 3a). The 10-years moving average (Fig. 3b) showed
a decrease in the wind direction until the early 70s, indicating a counter-clockwise rotation. However, during the following
decades the wind direction mainly rotated clockwise, returning to NE component, and overcoming the wind direction values
for the first decades of this study.

Changes during the annual cycle showed a predominant N-NW component from March to June (Fig. 4, green colour).
However, the NE component (light blue colour) prevailed during recent years. Usually, Trade Winds (light blue colour) are
not present during the winter months. However, a change in the wind direction pattern was observed during the last two
decades (1998-2018). This is coincident with the lower variability observed during the Trade Winds months (July, August
and September) of a fairly marked NE component since year 2000. Previous years showed a mixture of blue and green
colours (Fig. 4). According to Table 1, the lowest and highest values of wind direction were reached during decades of 1998-
2007 and 1958-1967, respectively. Examining the slope values, wind direction rotated counter-clockwise during the 50s, 60s,
80s and 90s while it did it in the opposite direction during the remaining decades (Fig. 3b and Table 1).

Average wind direction during the central month of winter (Fig. 5a) and autumn (Fig. 5d) showed the largest variability. By
contrast, spring (Fig. 5b) and summer (Fig. 5c) presented a more uniform direction over time. This is corroborated by the
seasonal statistical analysis included in Table 2. Quite low positive slopes were observed during all seasons, indicating that
wind direction remained fairly stable over the 70 years period, with a maximum slope of 0.288˚ showing a total variation of
less than 20º. Spring and summer displayed the lowest values (total variations of 3.27º and 6.56º, respectively), indicating
that Trade Winds hardly suffered a slight clockwise rotation during the seven decades analysed.

Wind intensity showed a linear regression with a slope of -0.0005 m·s$^{-1}$ for entire period, implying a small decrease of -0.42
m·s$^{-1}$ during the 70 years (Fig. 6a). The average wind intensity value was 5.1 m·s$^{-1}$ ± 2 m·s$^{-1}$ SD. The highest values of the
monthly wind intensity were observed during the 60s (10.4 m·s$^{-1}$ during July 1961 and 9.91 m·s$^{-1}$ in August 1962), while the
lowest were found in December 1961 and October 2015 (0.44 m·s$^{-1}$ and 0.46 m·s$^{-1}$, respectively). The 10-years moving
average of wind intensity (Fig. 6b) showed a sharp increase until 1963 and, mainly, a subsequent decrease until 2006.
Besides, another persistent increase was observed during the last decade. This behaviour is, as well, clearly observed in the
seasonal variability over the 70 years period (Fig. 7). Higher values were observed along the 60s during the Trade Wind
season (also in spring), decreasing during the next decades and slightly increasing again during this century (see also Table
3).

Regarding the seasonal study, time series during the central months of each season (Fig. 8) showed the most intense winds
occurring during summer (Fig. 8c) and spring (Fig. 8b) as expected. The lowest intensity was found during autumn (Fig. 8d)





and winter (Fig. 8a). The seasons displaying the largest intensity (spring and summer) showed sharp negative trends (see also Table 4), indicating that Trade Winds suffered a decrease in intensity of around 1 m·s$^{-1}$ over the 70 years period analysed.

Finally, we compared our wind time series with climatic indices such as the North Atlantic Oscillation (NAO) and the Atlantic Multidecadal Oscillation (AMO, Enfield et al., 2001). We found a significant relationship between wind intensity ($r^2 = 0.45$, $p < 0.05$) and direction ($r^2 = 0.27$, $p < 0.05$) with the NAO index during winter (Fig. 9), suggesting a close relationship with the general atmospheric circulation. We also found a significant relationship between the wind direction and the AMO index ($r = 0.61$, $p < 0,05$, Fig. 10a). However, no correlation was observed between the wind intensity and this oceanic climatic index (Fig. 10b). An increase of the intensity was observed after prolonged positive values of the AMO index, as observed during the 50s and 60s as well as during the present century. The correlation between the wind intensity and the AMO index changed quantitatively, and progressively increased showing the highest values for a lag over 10 years (Fig. 10b).

## 4. Discussion

In this study, we thoroughly analysed the meridional and zonal components of the wind north of the Canary Islands for the period from 1948 to 2017 using the NCEP/NCAR database with the aim of visualize changes in the wind patterns. Over the last 70 years, the wind direction suffered slight oscillations, rotating in a counter-clockwise direction several decades but changing to clockwise in the last ones. Regarding wind intensity, although the total net variation was small, major changes were observed with regular increases during the 50s and the last decade, decreasing during the other periods. We also found a significant correlation between wind direction and the AMO index for the entire period of study. However, the results of the correlation analysis between wind intensity and the AMO index seemed more complex. Bonino et al. (2019) observed different climatic features related to the different EBUS worldwide. Specifically, they found a negative upwelling trend connected to the low frequency of the AMO index in the Canary Current System. We also observed this relationship for the wind direction (Fig. 10a), but not for the wind intensity (Fig. 10b). The latter increased after prolonged warming (high AMO index) and decreased after prolonged cooling (low AMO index) over the Atlantic Ocean. We found significant correlations between wind intensity and the AMO index when the time lag between both parameters was longer than 10 years (Fig. 10b). This is similar to the lag found between the Atlantic multidecadal variability of sea surface temperature and surface turbulent heat fluxes observed by Gulev et al. (2013). This result suggests that ocean contributes to the multidecadal atmospheric variability with a considerable lag.

Our results also show a high correlation between wind (direction and intensity) and the NAO index, most remarkable when analysing seasonality, and displaying a widespread relationship for winter months throughout the period of study (1950–2017). This is due to the intensity of the day-to-day (synoptic) activity in the North Atlantic mid-latitudes which is closely linked to the North Atlantic Oscillation, strongly correlated with surface fluxes on short interannual to intra-decadal timescales (Hurrell et al., 1995; Eden & Willebrand, 2001; Gulev et al., 2013). This is an expected result as the variability of





wind across the Canarian Archipelago is known to be strongly influenced by the NAO (Hurrell et al., 2010; Häkkinen et al.,
2011; Barton et al., 2013; Cropper et al., 2014; Azorín-Molina et al., 2018).

Finally, changes in climatic indices such as wind intensity and direction could be the result of changes in the AMO index
rather than a simple decreasing trend. Climate in this area of the ocean seemed to be related to the large-scale variability of
the Atlantic Ocean rather than a local difference of pressure between the African continent and the Canary Current. The wind
variability appeared associated with shifts in the seasonal and interannual development and geographic positioning of the
four major atmospheric high-pressure systems. However, it is suggested that it is not directly promoted by the increase in the
local land-sea temperature difference associated with anthropogenic climate change as hypothesized by Bakun (1990).

In summary, during the last seven decades, the wind direction has experienced a slight increase with a net trend to rotate
clockwise 10.7°, while the intensity achieved a net decrease of 0.42 m·s$^{-1}$. However, it is important to emphasize that Trade
Winds were quite stable in direction, but they suffered a significant decrease in intensity of 1 m·s-1 along the 70 years,
although with some intensification during the last 15 years.  On the other hand, we found significant correlations between
NAO index and wind (direction and intensity), specifically in winter, indicating that the Canarian Archipelago is strongly
influenced by the NAO. We also found a significant correlation between wind direction and the AMO index for the entire
period of study. However, the correlation between wind intensity and the AMO index was found with a lag among both
parameters longer than 10 years. This result suggests that the ocean contributes to the multidecadal atmospheric variability
with a considerable time lag. In conclusion, changes in the wind patterns in the Canary Archipelago seemed to be related to
the large-scale variability of the Atlantic Ocean and not to local changes as hypothesized by Bakun.

**Data Availability**

The NCEP data were collected and made freely available by the Physical Sciences Division (PSD) of the National Oceanic
and Atmospheric Administration (NOAA) (https://www.esrl.noaa.gov/psd/). Last access: 17 March 2020.
The PODAAC data derived from the Cross-Calibrated Multi-Platform (CCMP) (https://doi.org/10.5067/CCF35-01AM1.)
Last access: 17 March 2020.
The AMO Index Data is calculated at NOAA/ESRL/PSD1 and available at
https://www.esrl.noaa.gov/psd/data/timeseries/AMO/ (last access: 17 March 2020)
The NAO Index Data is calculated at NOAA and available at https://www.ncdc.noaa.gov/teleconnections/nao/ (last access:
17 March 2020)

**Autor contributions**

SHL, DRE and JM contributed to design the research, review the literature and analyse the results. NMB was responsible for
the data processing and the generation of a first draft. All the authors were involved in the manuscript's discussion and
revision.





**Competing interests**

The authors declare that they have no conflict of interest

**Financial support**

This work has been supported by the PASTOR (ProID2017010072) project, funded by the Department of Economy, Industry, Commerce and Knowledge of the Canary Islands Government, and the European Fondo Europeo de Desarrollo
Regional (FEDER).

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





**Table 1. Statistical analysis of wind direction (degrees). Values are the result of the decadal study using monthly wind data from 1948 to 2017.**

|  | 1948-1957 | 1958-1967 | 1968-1977 | 1978-1987 | 1988-1997 | 1998-2007 | 2008-2017 |
|---|---|---|---|---|---|---|---|
| MIN | 54,65 | 70,05 | 14,73 | 57,68 | 65,21 | 11,37 | 46,48 |
| MAX | 307,72 | 338,41 | 241,97 | 265,18 | 284,1 | 281,6 | 298,46 |
| MEAN | 199,17 | 195,4 | 187,9 | 194,61 | 192,93 | 204,74 | 205,68 |
| SD | 26,26 | 31,31 | 34,91 | 32,24 | 36,11 | 28,84 | 33,79 |
| SLOPE | -0,1357 | -0,0022 | 0,1236 | -0,0248 | -0,1673 | 0,018 | 0,2318 |
| OFFSET | 207,38 | 195,53 | 190,42 | 196,11 | 203,06 | 203,65 | 196,66 |

**Table 2. Statistical analysis of wind direction (degrees) for the central month of each season (February for winter, May for spring, August for summer and November for autumn). Monthly wind data from 1948 to 2017 has been used.**

|  | FEBRUARY | MAY | AUGUST | NOVEMBER |
|---|---|---|---|---|
| MIN | 66,1 | 126,7 | 194,5 | 54,6 |
| MAX | 338,4 | 207,6 | 214,1 | 266,5 |
| MEAN | 198 | 190,6 | 202,3 | 203,6 |
| SD | 43,9 | 12,6 | 3,8 | 39,3 |
| SLOPE | 0,2988 | 0,0467 | 0,0938 | 0,1194 |
| OFFSET | 187,42 | 188,96 | 198,93 | 192,32 |

**Table 3. Statistical analysis of wind intensity (m·s⁻¹). Values are the result of the decadal study using monthly wind data from 1948 to 2017.**

|  | 1948-1957 | 1958-1967 | 1968-1977 | 1978-1987 | 1988-1997 | 1998-2007 | 2008-2017 |
|---|---|---|---|---|---|---|---|
| MIN | 0,98 | 0,44 | 0,81 | 1,2 | 0,55 | 0,72 | 0,46 |
| MAX | 9,03 | 10,43 | 8,64 | 9,03 | 8,25 | 8,61 | 8,9 |
| MEAN | 5,01 | 5,75 | 4,99 | 4,99 | 4,81 | 4,84 | 5,1 |
| SD | 1,88 | 2,18 | 1,84 | 1,91 | 1,84 | 1,92 | 2,05 |
| SLOPE | 0,0018 | -0,006 | -0,0006 | -0,0062 | -0,002 | -0,007 | 0,0064 |
| OFFSET | 4,90 | 6,11 | 5,03 | 5,36 | 4,94 | 5,26 | 4,72 |


**Table 4. Statistical analysis of wind intensity (m·s-1) for the central month of each season (February for winter, May for spring, August for summer and November for autumn). Monthly wind data from 1948 to 2017 has been used.**

|  | FEBRUARY | MAY | AUGUST | NOVEMBER |
|---|---|---|---|---|
| MIN | 0,8 | 2 | 4,1 | 0,6 |
| MAX | 7,9 | 8,1 | 9,9 | 6,8 |
| MEAN | 4,2 | 5,5 | 7,3 | 3,7 |
| SD | 1,7 | 1,4 | 1 | 1,4 |
| SLOPE | 0,0151 | -0,0158 | -0,0144 | 0,002 |
| OFFSET | 3,6281 | 6,0206 | 7,8272 | 3,6573 |




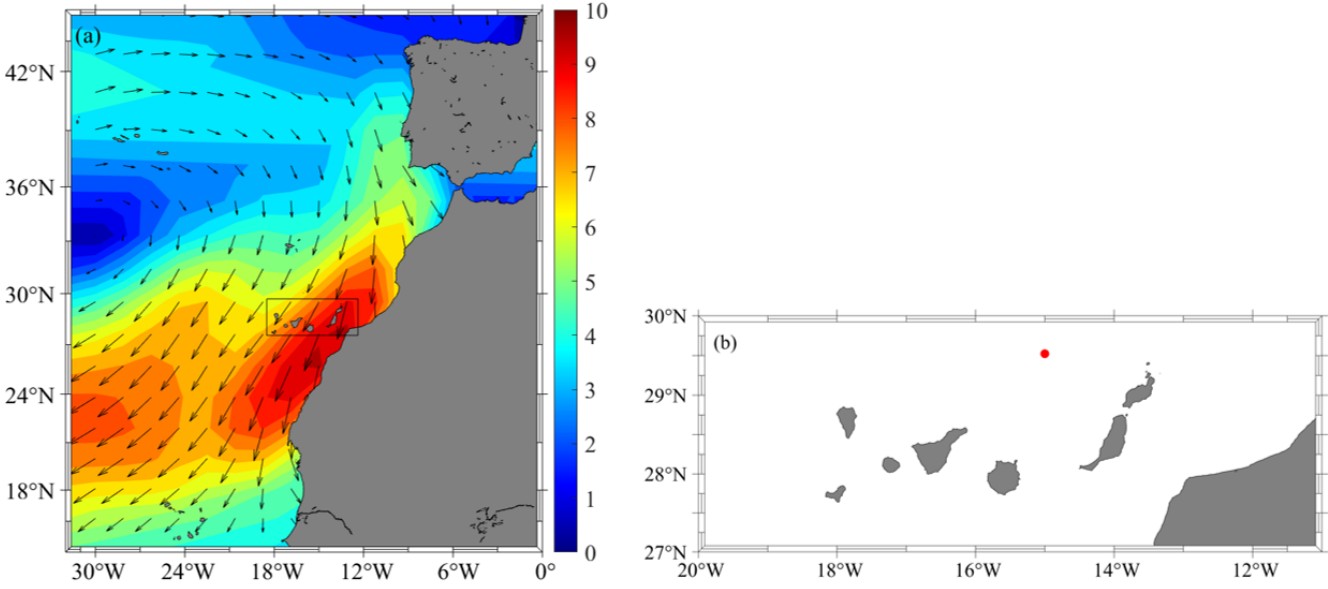

**Figure 1. (a) Geographic area of the Macaronesia during the Trade Winds season. Inset shows the study zone. The wind intensity (m·s⁻¹) is coloured and the wind direction is represented by the arrows. (b) Canary archipelago (the red dot represents the geographical location of the study area).**









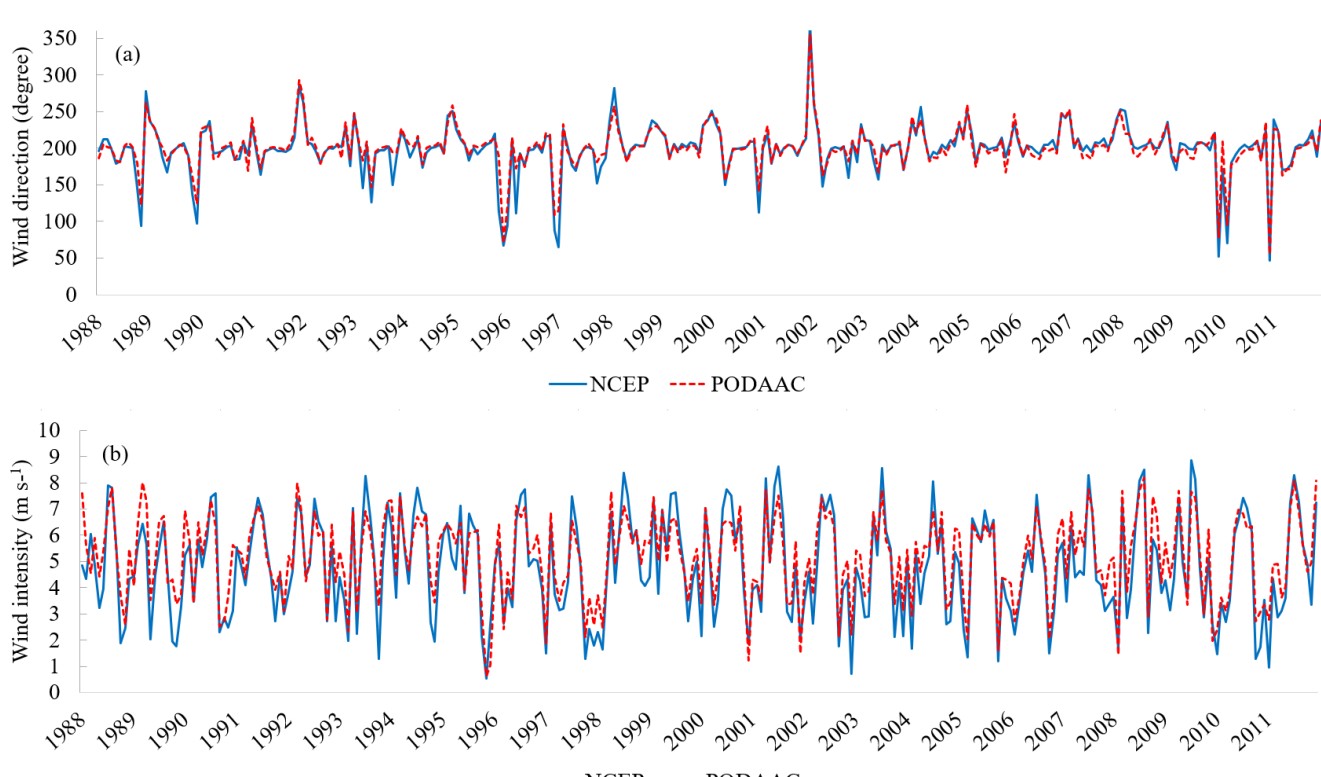

Figure 2. Correlation between NCEP and PODAAC. (a) Wind direction (degree). (b) Wind intensity (m·s⁻¹).



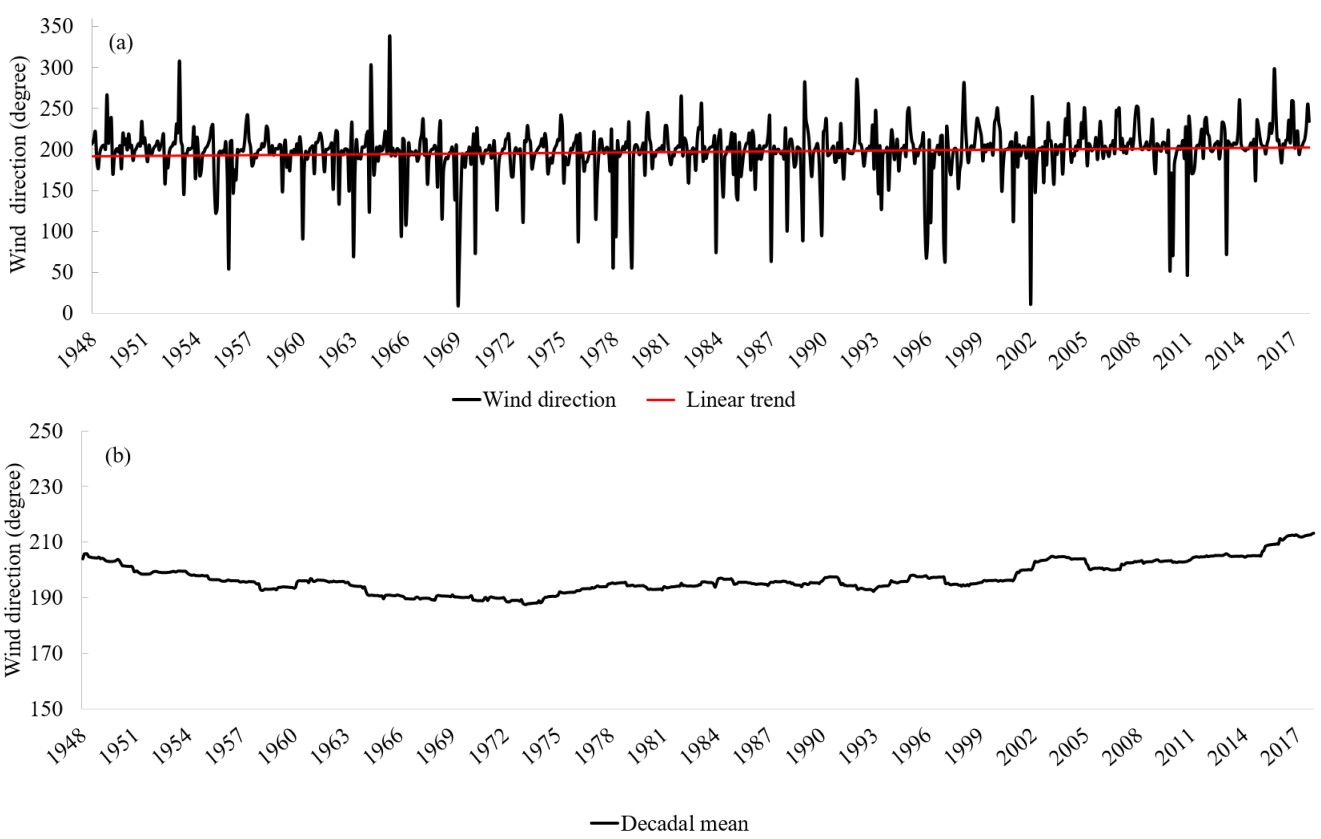

**Figure 3. Time series of (a) the wind direction with its linear regression, and (b) the 10-years moving average of the wind direction.**






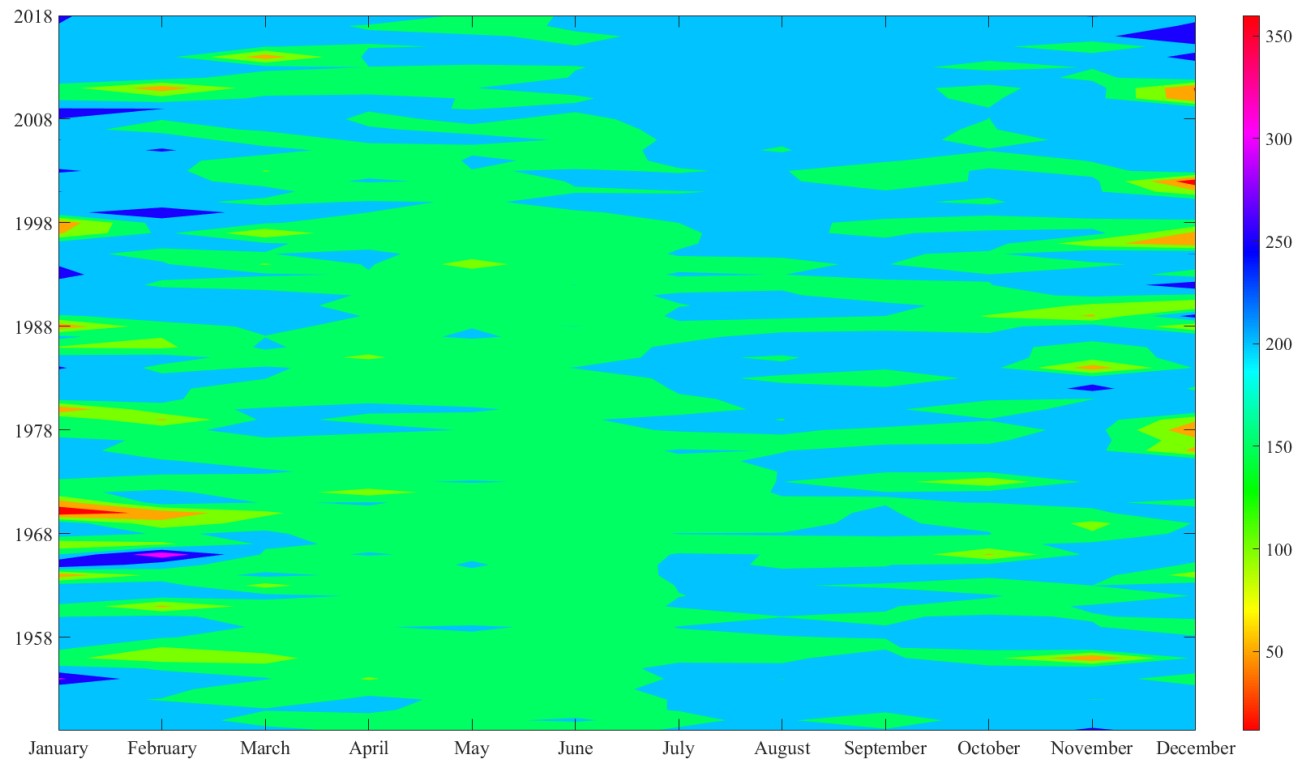

**Figure 4. Monthly variations of the wind direction (x axis) over the years (y axis). The palette is polar, that is, it begins and ends in the same color, so that the angular variation between 0° and 360° is intuitively appreciated.**







**Figure 5. Times series of the wind direction for the central month of each season, with their respective linear regression. Monthly data from 1948 to 2017. (a) Winter (February). (b) Spring (May).  (c) Summer (August). (d) Autumn (November).**

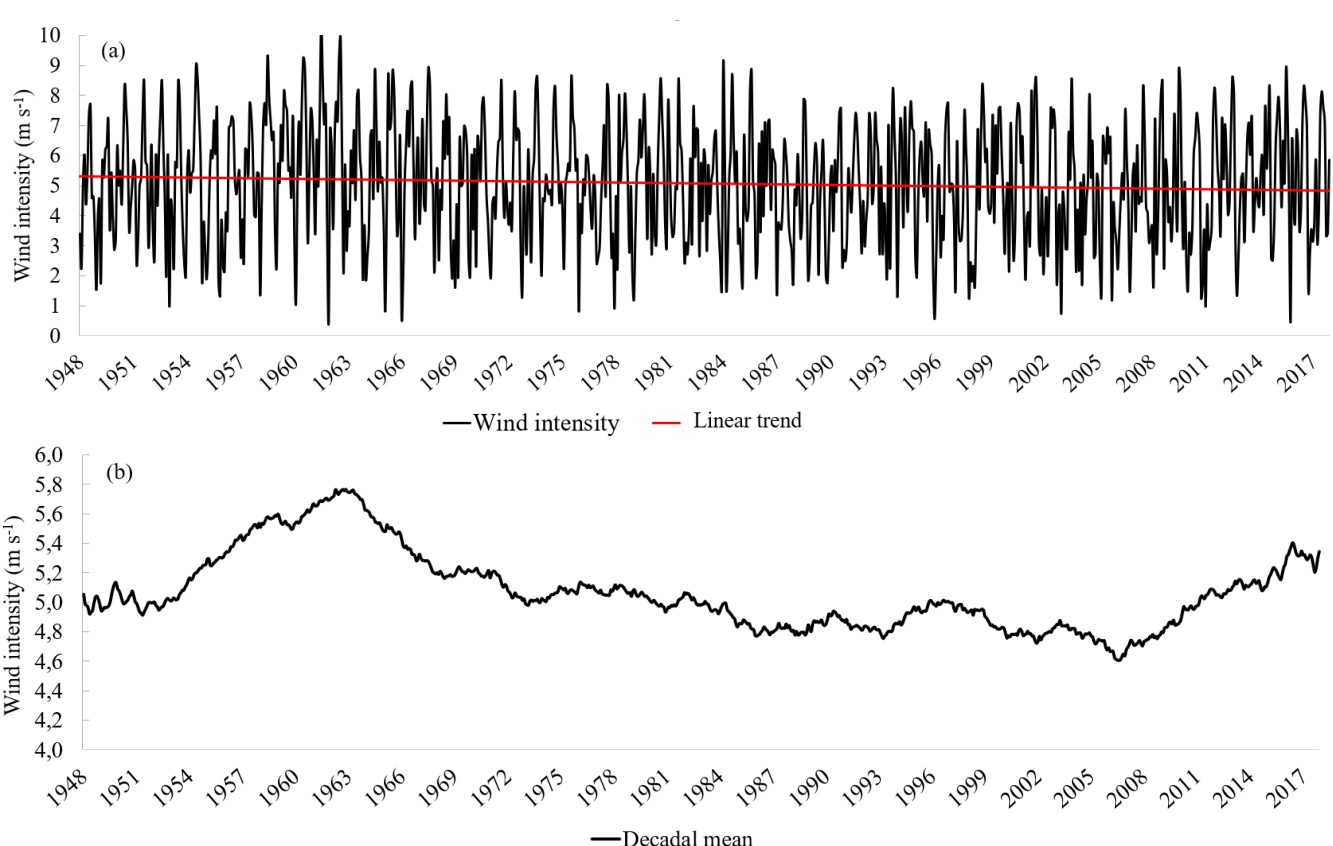

**Figure 6.** Time series of **(a)** the wind intensity with its linear regression, and **(b)** the 10-years moving average of the wind intensity.






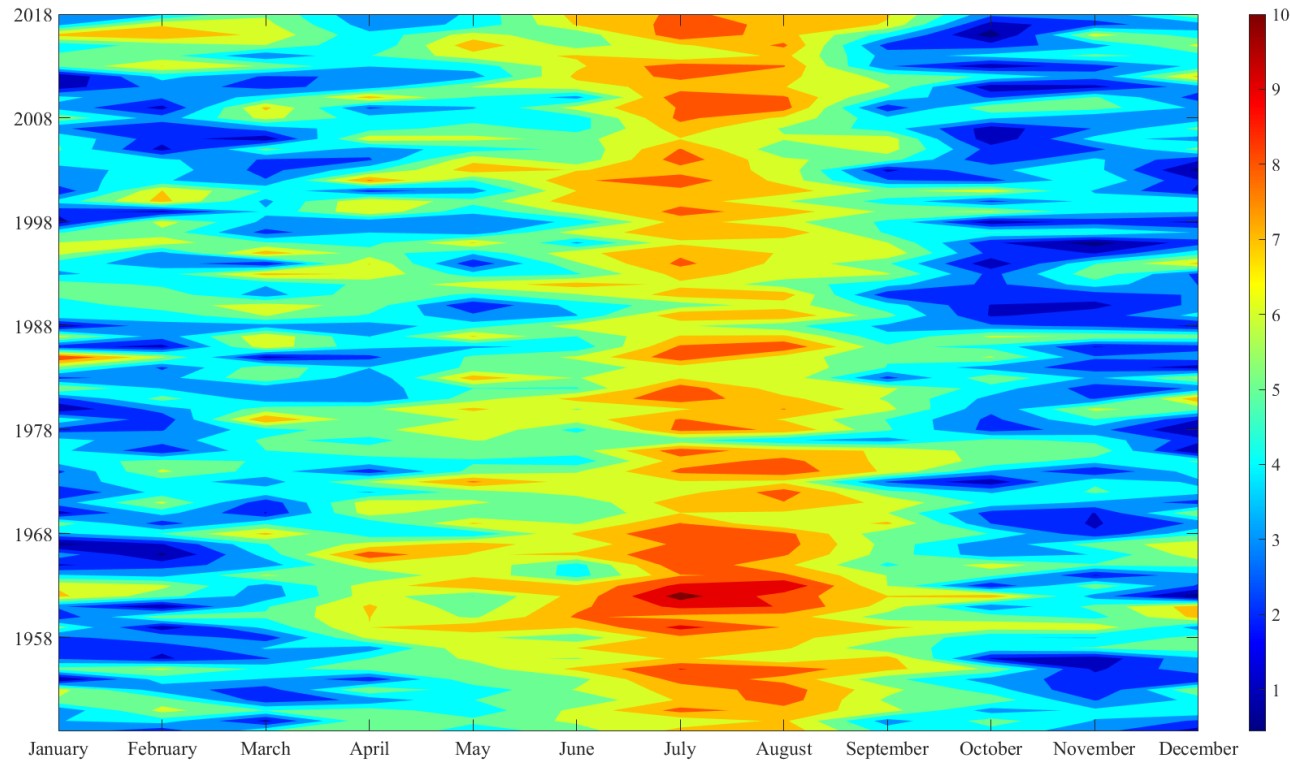

**Figure 7. Monthly variations of the wind intensity (m·s⁻¹) (x axis) over the years (y axis).**



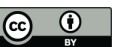




**Figure 8. Times series of the wind intensity for the central month of each season with their respective linear regression Monthly data from 1948 to 2017. (a) Winter (February), (b) Spring (May), (c) Summer (August), (d) Autumn (November).**








**Figure 9. Correlation time series between wind and NAO, from 1950 to 2017. (a) Times series of standardized wind intensity and NAO index in Winter (February) (b) Times series of standardized wind direction and NAO index in Winter (February).**



**Figure 10. Correlation time series between wind and AMO, from 1948 to 2014. (a) Time series of standardized decadal mean wind direction (left axis) and AMO index (right axis). (b) Time series of standardized decadal mean wind intensity (left axis) and AMO index (right axis). Inset shows the correlation with different time lags.**