# Peer review of "Wind variability in the Canary Current during the last 70 years"

_Ocean Science, 2020_

## Referee Comment (RC1) · Anonymous Referee #1 · 9 Apr 2020

"Wind variability in the Canary Current during the last 70 years"

The manuscript under review is eminently descriptive. It is focused on the changes of direction and intensity of the wind in a limited area of the Canary upwelling system. Anyway, it could be an interesting contribution to the field. At this point, the article should be upgraded in some ways to be suitable for publication. The paper presents some uncertainties and mistakes:

- Minor revisions:

1) Regarding the comparison of the databases, authors used PODAAC data which is limited from 1988 to 2011. There exists other database with similar resolution as, for example, CFSR which covers a temporal resolution from 1982 to nowadays. So author

should explain why they chose PODAAC being able to choose other options.

2) The selection of the area under study raises some uncertainties. First, the selection of the point north of the Canary Islands in Figure 1b. It is well-known that upwelling in the area occurs mainly near the coast and that weakens away from the coast. If the study is focused in the upwelling area and on the Bakun hypothesis, perhaps it should be better to choose points closer to the coast. Second, the area under study was set between 27oN and 30oN, and 11oW to 20oW. However, the Canary upwelling system covers a larger area. In fact, Cropper et al., (2014) established 3 different areas of the Canary upwelling system. The present study was limited to the "weak permanent upwelling" area of that study. Figure 1a shows that the highest wind intensity reach 24 oN in the south, why the authors limited their study to a smaller region? Is the north of the Canary Island the best spot to conduct studies about upwelling in the area?

3) The comparison with AMO and NAO raise some questions. To what extent are such low values of r (for example 0.45 or 0.27) sufficient to assume a certain relationship between the variables (Line 126)? Moreover, those values refer to February when upwelling is weaker or inexistent. To what extent these results affect upwelling in the area? Is there any explanation to establish a lag of 10 years (even closer to 20 years according to figure 10) or is it totally arbitrary?

4) It would be interesting to add a figure with a compass rose to show the seasonal behavior of the wind in a much more intuitive way.

- Minor comments:

1) Line 29: "Currently, the research about the response of the EBCS and the associated impact under a global climate change scenario have motivated numerous studies over different time periods" This statement should be supported by some references

2) The terminology "decrease or increase" of wind direction is controversial and more appropriated to wind intensity (for example in Line 94) . I recommend the authors to

use other expressions as "change"

3) Line 12: Delete "of the oceans"

4) Line 13: Change "word" to "world"

5) Lines 14-15: Change "Nearly 30 years ago, Bakun raised a hypothesis contending that coastal upwelling in eastern boundary upwelling systems (EBUS) might be intensified as the effect of global warming due to the enhancement of the Trade Winds as the effect of increasing pressure differences between the ocean and the continent" to "Nearly 30 years ago, Bakun raised a hypothesis contending that coastal upwelling in eastern boundary upwelling systems (EBUS) might be intensified by global warming due to the enhancement of the Trade Winds increasing pressure differences between the ocean and the continent".

6) Line 17: Change "theses" to "these"

7) Line 26: Delete "of the world ocean"

8) Figure 5, 8 and 9: Change "times" to "time"

9) References: - Some references are overlapped - Debernard and Roed, 2008 does not appear in the text - Hurrell (both references) are wrong in the text (Lines 153-154)

Please also note the supplement to this comment:
https://www.ocean-sci-discuss.net/os-2020-18/os-2020-18-RC1-supplement.pdf

---

## Author Comment (AC1) · 25 Apr 2020

1) Regarding the comparison of the databases, authors used PODAAC data which is limited from 1988 to 2011. There exists other database with similar resolution as, for example, CFSR which covers a temporal resolution from 1982 to nowadays. So author should explain why they chose PODAAC being able to choose other options.

Thanks for the comment. We selected the PODAAC database as the initial approach to the wind data in the region, but in order to cover a longer time-series we correlated the data with the NCEP/NCAR database, and we obtained statistically significant correlation results (r=0.949, r=0.923, wind direction and wind intensity, respectively). Thus, we decided to make the study with the NCEP database. The length of the NCEP/NCAR

time-series allowed to conduct a study on wind patterns covering a temporal slot still not addressed in the scientific literature for the Canary Current, as far as we know. Certainly, CFSR is another data source and it is a reanalysis product that assimilates satellite radiances, but the most important aspect to our view was to cover the longest temporal coverage, and therefore the optimal choice was NCEP.

Action:

To clarify this issue, we included additional text in section "3. Results", line 92. In addition, we added the corresponding references in the References section (line 295).

2) The selection of the area under study raises some uncertainties. First, the selection of the point north of the Canary Islands in Figure 1b. It is well-known that upwelling in the area occurs mainly near the coast and that weakens away from the coast. If the study is focused in the upwelling area and on the Bakun hypothesis, perhaps it should be better to choose points closer to the coast. Second, the area under study was set between 27oN and 30oN, and 11oW to 20oW. However, the Canary upwelling system covers a larger area. In fact, Cropper et al., (2014) established 3 different areas of the Canary upwelling system. The present study was limited to the "weak permanente upwelling" area of that study. Figure 1a shows that the highest wind intensity reach 24 oN in the south, why the authors limited their study to a smaller region? Is the north of the Canary Island the best spot to conduct studies about upwelling in the area?

Thanks for your comments. Our study aimed to analyze the wind pattern in the Canary Current over the last 70 years. In particular, the study mainly addressed the analysis of the Trade Winds in the area. These prevailing regular winds decisively influence the climate of the region. However, as they blow from the North-East direction, we selected the most suitable location to analyze its effects over time. Locations to the south are quite disturbed by the islands all over hundreds of kilometers. On the other hand, sampling sites closer to the African continent were not chosen as winds near the coast behave differently as they are affected by the day-night cycle. Because of

the difference in temperature between the ocean and the desert, the coastal zone is noisier than the proper ocean. An example is the study by Grall et al. (1982) who observed an important variability in daily wind intensity, also affecting primary production. Differences in wind intensity between the coastal zone (ICOADS data) and the ocean (WASWind data) were not statistically different but highly correlated in the study by Barton et al. (2013). Therefore, the area selected for the present study seemed to our view the most suitable to characterize wind patterns in the area. Moreover, the coincidence of our data with the European Oceanic Time-Series Station (ESTOC), where water column temperature exists from the 90s will allow a future comparison among both data sets. In any case, the upwelling intensity is related to the general pattern of wind direction and intensity in the area (Ekman transport), and not specifically related to the variable coastal phenomena mostly related to topography.

Action: To clarify this issue, we included additional text in section "2.1 Study Area", Line 57.

3) The comparison with AMO and NAO raise some questions. To what extent are such low values of r (for example 0.45 or 0.27) sufficient to assume a certain relationship between the variables (Line 126)? Moreover, those values refer to February when upwelling is weaker or inexistent. To what extent these results affect upwelling in the area? Is there any explanation to establish a lag of 10 years (even closer to 20 years according to figure 10) or is it totally arbitrary?

The reviewer is right. We found a significant relationship between the NAO index and wind direction and intensity, but it only reflects a general pattern related to the general atmospheric circulation, and not a close relationship as stated in the previous version. In relation to the lag observed between the AMO index and the wind intensity (but not direction), we cite in the manuscript the paper by Gulev et al. (2013) who found also a considerable lag between the Atlantic Multidecadal Variability and surface turbulent fluxes. They discuss the suggestion by Bjerknes (1964) that the atmosphere drives short-term (interannual) sea surface variability, and the ocean contribute to long-term

(multidecadal) sea surface temperature and potentially atmospheric variability. Gulev et al. (2013) observed surface turbulent heat fluxes driven by the ocean and forcing the atmosphere on times scales longer than 10 years. We think that there is no scope to this study to go in depth to this general problem but we show some agreement with the results of Gulev et al. (2013). We think it is worthy to discuss in brief.

Action: To clarify this issue, we included additional text in section "4. Discussion", line 154, and we edit the line 179. In addition, we added the corresponding references in the References section (line 231).

4) It would be interesting to add a figure with a compass rose to show the seasonal behavior of the wind in a much more intuitive way.

Thanks for your suggestion. As requested, we produced a wind rose for each season of the year and included it in the manuscript for better understanding and visualization; However, we decided to leave the time series of wind direction and intensity, as the trends are more visual in these figures.

Action: We added the new Figure 6 in the manuscript (Line 385) and we have referenced it in the main text accordingly.

Minor comments: 1) Line 29: "Currently, the research about the response of the EBCS and the associated impact under a global climate change scenario have motivated numerous studies over different time periods" This statement should be supported by some references

Agreed. Four references were included to support the statement.

2) The terminology "decrease or increase" of wind direction is controversial and more appropriated to wind intensity (for example in Line 94) . I recommend the authors to use other expressions as "change"

Agreed. Text amended accordingly.

3) Line 12: Delete "of the oceans"

Agreed. The text deleted.

4) Line 13: Change "word" to "world"

Agreed. Done

5) Lines 14-15: Change "Nearly 30 years ago, Bakun raised a hypothesis contend-ingthat coastal upwelling in eastern boundary upwelling systems (EBUS) might be inten-sified as the effect of global warming due to the enhancement of the Trade Winds asthe effect of increasing pressure differences between the ocean and the continent" to"Nearly 30 years ago, Bakun raised a hypothesis contending that coastal upwelling ineastern boundary upwelling systems (EBUS) might be intensified by global warming-due to the enhancement of the Trade Winds increasing pressure differences between-the ocean and the continent".

Agreed. The paragraph is now replaced

6) Line 17: Change "theses" to "these"

Thanks. Done.

7) Line 26: Delete "of the world ocean"

Thanks. Words were deleted.

8) Figure 5, 8 and 9: Change "times" to "time" Agreed. Done.

9) References: - Some references are overlapped - Debernard and Roed, 2008 does not appear in the text - Hurrell (both references) are wrong in the text (Lines 153-154)

Thanks. References are now corrected.

Please also note the supplement to this comment:
https://www.ocean-sci-discuss.net/os-2020-18/os-2020-18-AC1-supplement.pdf

---

## Referee Comment (RC2) · Anonymous Referee #1 · 4 May 2020

Thank you for your answers. I consider that all the doubts have been answered correctly, the changes and additions are significant and the reviewer's comments have been taken into account. I consider that the article should be considered for publication

---

## Referee Comment (RC3) · Anonymous Referee #2 · 16 May 2020

General CommentsÂă: The authors have tried to analyze the long term trend of the wind intensity with respect to the climatics indices mainly NAO and AMO.

In this short paper, various analysis have been done in various ways to investigate factors controlling the wind changes patterns along the Canary Archipelagos. There may be interesting findings, it discusses an aspect which is already very controversial, it is the Bakun hypothesis which stipulates an acceleration of the wind and intensifies the upwelling.

Main comments:

Regardless of whether the paper is accepted or not, I have two concerns:

1- The use of CCMP data is very doubtful since the time series before 2009 date of the

injection of QUIKSCAT data (1999-2009) in the variational analysis underestimates the intensity of the winds and overestimates the wind after 1999. In this context, and to be able to conclude on the trend, the authors have to use other data sources, especially ASCAT, although the series is short, but it allows to validate or correct the data series.

2- The use of such a long time series would undoubtedly reveal a periodicity in the variability of the wind, could you calculate or estimate the duration of the periods of high intensities and those of low intensities.

Please also note the supplement to this comment:
https://www.ocean-sci-discuss.net/os-2020-18/os-2020-18-RC3-supplement.pdf

---

## Author Comment (AC2) · 4 Jun 2020

1)The use of CCMP data is very doubtful since the time series before 2009 date of the injection of QUIKSCAT data (1999-2009) in the variational analysis underestimates the intensity of the winds and overestimates the wind after 1999. In this context, and to be able to conclude on the trend, the authors have to use other data sources, especially ASCAT, although the series is short, but it allows to validate or correct the data series.

Thanks for the remark. We performed a correlation study between the CCMP V1.1 (PODAAC) database we used in our analysis and the new CCMP V2.0 database (released by REMSS in 2017). This reprocessing is an update of CCMP. It uses the most current and complete RSS cross-calibrated wind datasets, including ASCAT, and

uses the ECMWF Interim reanalysis as a consistent and higher resolution background. In addition, in CCMP V2.0 QuikSCAT wind data are improved when compared to the CCMP V1.1 winds. In any case, the main caution using the CCMP database is when studying high wind regions (wind speeds >25 m/s). As presented in the manuscript, in our region Trade Winds dominate and, therefore, wind intensity is usually moderate (around 5 m/s). In addition, problems with CCMP may arise in heavy rain scenarios; however, in our region these events are infrequent. The correlation between the CCMP V1.1 database and the new CCMP V2.0 database in our region for the wind intensity is 0.967 (Figure 1) and for the wind direction is 0.978 (Figure 2). Results demonstrate that both datasets are highly correlated in direction and intensity.

To clarify the work scope, our study was performed only using the NCEP data due to the long period of time covered. We just used the PODAAC database to validate the reliability of the NCEP reanalysis. In this context, now we performed, as well, the correlation study between the new CCMP V2 database and the NCEP database. The correlation value for the wind intensity is 0.964 (Figure 3) and for the wind direction is 0.972 (Figure 4). The study covered the 24 years period (from 1988 to 2011), as in our manuscript.

In summary, as shown in Figures 1 and 2, the correlation between PO-DAAC and the improved version is very high and correlations with NCEP are very similar. Therefore, the CCMP V1 in our region provides accurate wind estimates. The web links to the PODAAC and REMSS databases are: https://podaac.jpl.nasa.gov/dataset/CCMP_MEASURES_ATLAS_L4_OW_L3_5A_MONTHLY_WIND_VECTORS_FLK and http://www.remss.com/measurements/ccmp/.

Action: Even though PODAAC and CCMP V.2 have high correlation, to clarify this issue and to include the most updated wind data, we replaced the PODAAC database by CCMP V.2 data in the manuscript. We also extended the temporal coverage (1988-2017). Specifically, we included additional text in section "2.2. Data" (line 70), and we added the corresponding references in the References section (lines 216, 220 and

274). In addition, we changed the old Figure 2 to new Figure 2 with CCMP V2 data (line 390).

2)The use of such a long time series would undoubtedly reveal a periodicity in the variability of the wind, could you calculate or estimate the duration of the periods of high intensities and those of low intensities.

Thanks for the comment. With respect to the periodicity of the wind, we performed a Fourier and a Wavelet analysis to examine the wind intensity periodicity. As expected, the results of the Fourier analysis (Figure 5) show that there is a predominant annual periodic pattern; however, but with much lower magnitude six-monthly and four-month patterns appear as well.

These results are also confirmed by the continuous Wavelets analysis (Figure 6), where the annual pattern is clearly present (x-axis value for the horizontal yellow line) in the wind intensity. In addition, this pattern seems to be discontinuous through the entire time series as it reveals the change in magnitude observed in Figure 6, where the annual periodicity moves its colour from orange/yellow to green/blue during some periods.

Regarding the periods of high and low intensities, firstly, we would like to indicate that the average wind intensity value for the 70 years' period is 5.1 m/s $\pm$ 2 m/s SD. Table 3 (of the manuscript) shows the mean, maximum and minimum values of the wind intensity for each decade. Figure 7 shows the behaviour of wind intensity over the 7 decades analysed representing the decadal mean of the intensity (black line) and the average value of wind intensity for the complete period (red line). Specifically, the highest values of the wind intensity were observed during the 50s and mainly during the 60s (10.4 m/s during July 1961 and 9.91 m/s in August 1962). During the 70's the wind intensity remained stable. Then, in the following decades (80s, 90s and 2000s) the wind intensity was lower than in previous decades. Finally, in recent years, the intensity of the wind is increasing again.

Action: To clarify this issue, we included additional text in section "3. Results" (lines

107 and 127), and we improved Figure 4 (line 410), and Figure 8 (line 458), adding the wavelet analysis to show the periodicities along the years.

Please also note the supplement to this comment:
https://www.ocean-sci-discuss.net/os-2020-18/os-2020-18-AC2-supplement.pdf
* * *
[Figure]

[Figure]

[Figure]

[Figure]

[Figure]

[Figure]

[Figure]

[Figure]

Fig. 7.

(b)

Wind intensity (m s⁻¹)

—Decadal mean

---

## Author Comment (AC3) · 5 Jun 2020

Thank you very much.

———————————————————